# Preclinical Evaluation of a Microwave-Based Accessory Device for Colonoscopy in an In Vivo Porcine Model with Colorectal Polyps

**DOI:** 10.3390/cancers15123122

**Published:** 2023-06-08

**Authors:** Alejandra Garrido, Marta Guardiola, Luz María Neira, Roberto Sont, Henry Córdova, Miriam Cuatrecasas, Krzysztof Flisikowski, Joel Troya, Josep Sanahuja, Thomas Winogrodzki, Ignasi Belda, Alexander Meining, Glòria Fernández-Esparrach

**Affiliations:** 1MiWEndo Solutions S.L., 08014 Barcelona, Spain; agarrido@miwendo.com (A.G.); marta@miwendo.com (M.G.); lmneira@miwendo.com (L.M.N.); rsont@miwendo.com (R.S.); ibelda@miwendo.com (I.B.); 2Endoscopy Unit, Gastroenterology Department, Hospital Clínic, University of Barcelona, 08036 Barcelona, Spain; hcordova@clinic.cat; 3Biomedical Research Network on Hepatic and Digestive Diseases (CIBEREHD), 28029 Madrid, Spain; mcuatrec@clinic.cat; 4Institut d’Investigacions Biomèdiques August Pi i Sunyer (IDIBAPS), 08036 Barcelona, Spain; 5Facultat de Medicina i Ciències de la Salut, University of Barcelona, 08036 Barcelona, Spain; 6Pathology Department, Hospital Clínic, University of Barcelona, 08036 Barcelona, Spain; 7Lehrstuhl für Biotechnologie der Nutztiere, School of Life Sciences, Technische Universität München, 80333 München, Germany; krzysztof.flisikowski@tum.de (K.F.); thomas.winogrodzki@tum.de (T.W.); 8Interventional and Experimental Endoscopy (InExEn), Gastroenterology, Internal Medicine II, University Hospital Würzburg, 97080 Würzburg, Germany; troyasebas_j@ukw.de (J.T.); meining_a@ukw.de (A.M.); 9Anesthesiology Department, Hospital Clínic, University of Barcelona, 08036 Barcelona, Spain; sanahuja@clinic.cat

**Keywords:** colonoscopy, microwave imaging, microwave-based colonoscopy

## Abstract

**Simple Summary:**

Colonoscopy is the most effective method for colorectal cancer prevention but has several drawbacks that result in polyp miss rates. Microwave imaging is a novel solution that can obtain images of the colon based on the capacity of detecting changes in the dielectric properties of tissues. The aim of the current study is to evaluate, for the first time, the feasibility of a microwave-based accessory device for colonoscopy in an in vivo porcine model. We developed a device provided with microwave antennas and attached it to a conventional endoscope. In this preclinical model, the device did not impair endoscopic vision and was compatible with endoscopic tools; deep mural injuries were not observed and different signals were obtained from polyps and healthy mucosa, confirming its usability and potential of detecting polyps.

**Abstract:**

Background and Aims: Colonoscopy is currently the most effective way of detecting colorectal cancer and removing polyps, but it has some drawbacks and can miss up to 22% of polyps. Microwave imaging has the potential to provide a 360° view of the colon and addresses some of the limitations of conventional colonoscopy. This study evaluates the feasibility of a microwave-based colonoscopy in an in vivo porcine model. Methods: A prototype device with microwave antennas attached to a conventional endoscope was tested on four healthy pigs and three gene-targeted pigs with mutations in the adenomatous polyposis coli gene. The first four animals were used to evaluate safety and maneuverability and compatibility with endoscopic tools. The ability to detect polyps was tested in a series of three gene-targeted pigs. Results: the microwave-based device did not affect endoscopic vision or cause any adverse events such as deep mural injuries. The microwave system was stable during the procedures, and the detection algorithm showed a maximum detection signal for adenomas compared with healthy mucosa. Conclusions: Microwave-based colonoscopy is feasible and safe in a preclinical model, and it has the potential to improve polyp detection. Further investigations are required to assess the device’s efficacy in humans.

## 1. Introduction

With almost two million new cases and one million deaths globally in 2020, colorectal cancer (CRC) is the third-most diagnosed and the second-most lethal malignancy. Thus, CRC has become a substantial burden to global health, especially in countries with a Westernized lifestyle [1]. The tumor stage at the time of diagnosis plays a major role in patient survival. The overall 5-year survival of CRC patients is close to 65%; 5-year survival rates range from 90% for patients with localized disease to 70% and 13% for regional and distant stages, respectively. Furthermore, differently from many cancers, CRC can be prevented and potentially cured if early-stage tumors and high-risk adenomas are removed [2]. Thus, since CRC is mostly asymptomatic until it progresses to advanced stages, the implementation of screening programs for early detection is essential to reduce incidence and mortality rates.

Currently, colonoscopy, by identifying and removing polyps with good accuracy, is considered the most effective diagnostic and therapeutic technique for preventing colorectal cancer (CRC). Studies show that colonoscopy with polypectomy can reduce the incidence of colorectal cancer by 40–90% [2,3]. However, colonoscopy is not perfect as 22% of polyps may be undetected [4], and there is still a 7.9% risk of cancer after a negative colonoscopy [5]. These limitations arise due to the visualization issues of endoscopes, which can miss up to 13.4% of the colon’s surface area during a standard colonoscopy [6,7]. To overcome these issues, various devices and technologies have been developed, such as endoscopes with multiple lenses and mucosal flattening accessories [8]. Another limitation of colonoscopy is the different performances among endoscopists, resulting in an unacceptable variability in the adenoma detection rate (ADR) [9,10]. Therefore, a tool capable of automating the detection of polyps is needed. Artificial intelligence is increasingly used for the real-time assessment of endoscopic images, showing an increase in ADR of 14% [11]. The main drawback of this method is that it is limited by what is captured by the camera.

Microwave imaging is another potential and innovative solution that can generate images without field-of-view restrictions and can penetrate light opaque tissues with non-ionizing radiation [12,13]. It can obtain both anatomical and functional images of the colon’s interior, representing the contrast in dielectric properties between polyps and healthy colon mucosa [14]. The dielectric properties—relative permittivity and conductivity—are biomarkers of several health conditions, such as breast cancer [12], brain stroke, osteoporosis, heart infarction, and colorectal polyps and cancers [13]. MiWEndo Solutions (Barcelona, Spain) has developed the first microwave-based accessory device that can be attached at the end of a colonoscope. The accessory is connected to an external processor that generates microwaves and analyzes the retrieved signals, and it is programmed to produce an acoustic signal when a polyp is detected.

As an active and invasive device, strict regulatory requirements apply, such as biocompatibility, water tightness, electrical safety, electromagnetic compatibility, usability, and ergonomics, to avoid injuring the patient and affecting the maneuverability or changing the clinical practice. The manufacturing cost is also an important concern as colonoscopy accessories are recommended for single use. Finally, the overall system must have a real-time response to notify the doctor of the presence of a polyp at the same time as when the endoscope tip passes through the lesion. All these requirements have been considered as design inputs for the version of the prototype used in the present study.

The preclinical validation of a medical device consists of a series of increasingly realistic experiments with growing complexity. The device has recently demonstrated its capability of detecting lesions in a phantom simulating a colon [15] and in excised human colons with CRC or polyps [16]. The IDEAL framework suggests performing preclinical studies in animal models before implementing any innovation in patients [17]. The porcine model with polyps created by Flisikowska et al. [18] was demonstrated as suitable for the evaluation of endoscopic procedures, using human-sized equipment that is most suitable for testing medical devices [19]. In this study, we present the in vivo preclinical evaluation of a microwave-based colonoscopy using a new accessory medical device for detecting polyps as a complement to conventional endoscopic images. In Section 2, the materials and methods of the study are presented by describing the pig model used, the microwave-based device and method, and the study protocol. In Section 3, the results of the study are presented, and in Section 4, a discussion of the study’s findings and limitations is detailed.

## 2. Materials and Methods

### 2.1. Pig Model

Experiments were performed in 4 healthy female Yorkshire pigs and in 3 gene-targeted pigs with 1311 truncated mutations in the APC gene [18]. Animals were housed in controlled conventional environmental conditions at the animal facility of the School of Medicine, University of Barcelona, and School of Life Sciences, Technische Universität München. Colons were prepared with water enemas until a clean visualization was obtained. The distal colon and rectum of pigs were explored. Colonoscopy was conducted on the animals under anesthesia, and all procedures were performed by a single endoscopist. Immediately after the experiment, the 4 healthy pigs were sacrificed, and the other 3 were recovered and sent to their cages. During necropsy, the distal colon and rectum were excised and opened to look for mural injuries. Afterwards, the specimens were placed in formalin and sent to the Pathology Department. The studies were performed according to the Animal Research Committee of the University of Barcelona, the German Animal Welfare Act, and the European Union Normative for Care Use of Experimental Animals.

### 2.2. Microwave-Based Device Description

This current imaging system is an improved version of the previous one [16]. It comprises two main parts: (1) a cylindrical ring-shaped acquisition device that can be attached to the end of a standard colonoscope and (2) an external unit containing a microwave transceiver and a processing unit. The acquisition device is an encapsulated printed circuit board with two arrays of eight antennas, one for transmitting and the other for receiving [20]. These antennas are connected to the external unit by using cables (see Figure 1). The acquisition device measures 30 mm in length and 20 mm in diameter, and it has a total thickness of 2.5 mm.

### 2.3. Description of Microwave Imaging Method for Colonoscopy

The microwave imaging method for colonoscopy utilizes several antennas surrounding the colonoscope tip to emit microwaves and capture the resulting waves produced by the interaction with the colon. These antennas, operating at 7.5 GHz, are designed to cover the full perimeter of the colon. To obtain information for imaging purposes, the antennas are sequentially switched as follows: The total received field is measured at the receiving antenna adjacent to the active transmitting antenna and at the two closest diagonal antennas. This process is repeated for each transmitting antenna surrounding the colonoscope’s tip to obtain information on the entire colon or rectum perimeter. The received field includes information on the spatial changes in dielectric tissue properties, and by processing the field using an imaging algorithm, an image of the colorectal tissue’s dielectric property contrast can be obtained in real time. This image represents a cross-sectional slice of the colon or rectum, which is known as a frame. As the colonoscope moves, the acquisition device continuously measures the microwave field and obtains frames, covering the entire colorectal lumen surface. The frames that correspond to a healthy section of the colon have a homogeneous appearance, while when bright pixels appear, it indicates the presence of a polyp. The detection of the polyp can also be displayed as an acoustic signal.

### 2.4. Study Protocol

The distal colon and rectum of pigs were explored using the microwave-based device. In the 4 healthy pigs, safety and compatibility with endoscopic tools and the electrosurgical unit were assessed. The overall usability (including the assembly of the device, interferences with the optic, and maneuverability) of the current design was also assessed using a five-point Likert scale.

Thereafter, the detection capability of the microwave-based device was tested in 3 further gene-targeted pigs with adenomas. The endoscopic image and the microwave image obtained during the retrieval of the endoscope were recorded and stored. Using an external computer, the microwave image was timely correlated with the endoscopic image and used as the ground truth.

## 3. Results

In the initial animal trial, which involved four healthy pigs, colonoscopies were performed without causing any significant damage to the colon wall, as confirmed by histological analysis. The use of electrocautery and water instillation did not interfere with the device’s ability to emit and receive microwaves, and none of the device components were disassembled during the procedure. Usability scores of 4 or higher were given to all components.

During the second animal trial, which involved three pigs with mutations in the APC gene, a total of 7087 frames were recorded over a period of 15 min. However, a significant number of frames were not useful due to direct contact with the mucosa or stool remnants (14%). When the accessory touched the colon walls or stool remnants, the antennas were mismatched and lost their normal performance. Due to the pigs having a large number of polyps that were clustered together, many frames were discarded (59%). These frames were excluded as they were not a true representation of patients that were referred for a diagnostic or screening colonoscopy (Figure 2).

The remaining valid frames correspond to a sequence of 600 frames with only three isolated polyps found in one of the pigs. This sequence was processed with the microwave imaging algorithm to obtain the aggregated dielectric contrast of each frame. All frames of the sequence are concatenated to obtain the center image in Figure 3. Frames were grouped in segments containing polyps (*n* = 3), and these are represented in yellow; frames were also grouped in segments containing healthy mucosa (*n* = 4), and these are represented in blue in the bottom image in Figure 3. As observed in the center image, the system recognized all segments with polyps. In this image, detections, or higher contrast areas, are represented in bright yellow, while a healthy colon is represented in blue. True polyps are detected with wider bright areas, whereas shorter bright areas correspond to stool remnants.

## 4. Discussion

The present study showcases preclinical findings on a novel approach to colonoscopy utilizing microwave imaging for automatically detecting polyps. While microwave imaging has already demonstrated its diagnostic abilities in breast cancer research [21,22], this study has a novelty that combines, for the first time, endoscopic imaging with a qualitative measure of colonic tissue dielectric properties. For carrying this out, an accessory device that can be attached to the end of any conventional colonoscope in terms of size, allowing for differentiation between healthy mucosa and neoplastic lesions based on their dielectric properties, has been developed [15,20]. This technology offers an advantage by adding microwave vision to conventional colonoscopy’s optical vision.

Unlike other technologies that are limited to what is visible in the optical image, our microwave-based colonoscopy has the ability to distinguish between healthy mucosa and neo-plastic lesions by detecting changes in their dielectric properties, thereby complementing the endoscopic image. This device could prove particularly useful in identifying small flat adenomas and polyps hidden behind mucosal folds that often remain unnoticed. In a previous study, we showed that the dielectric properties are not dependent on the shape of polyps [14]. Polyps behind mucosa folds also constitute a large percentage of missed lesions due to the limited field of view of current endoscopes. Therefore, a device capable of scanning the mucosa along 360° could help overcome this problem.

Besides the potential improvement in the detection of polyps, the differences in the dielectric properties showed by the different types of polyps and degrees of dysplasia [14] could allow microwave-based colonoscopy to differentiate adenoma from cancer or adenoma from hyperplastic polyps in real-time explorations. The in vivo characterization of polyps and histological prediction are of paramount importance because they guide decisions that are clinically relevant, such as the decision to resect or not, the type of resection technique, or the need for delaying the polypectomy or referring the patient to a more expert center [23].

The system has been designed to maintain compatibility with colonoscopy, provide full 360° coverage, and minimize any changes to current clinical procedures. The final dimensions of the acquisition device are 30 mm in length by 20 mm in diameter, with a total thickness of 2.5 mm, in consideration of size restrictions. The shape and size of the device have been chosen to avoid blocking the front tip of the colonoscope, prevent injury to the patient, and enable the smooth maneuverability of the colonoscope. To validate the system’s performance, the device has been tested on ex vivo human colons with polyps and a colon phantom, which realistically simulates a colonoscopy examination, including the haustrum and placement of different polyps [15]. The phantom is composed of materials, such as gelatin, water, and oil, that replicate the dielectric properties of healthy colon mucosa and polyps with high-grade dysplasia [16]. However, to demonstrate the safety of the device and its compatibility with a real exploration, an in vivo experiment was needed. In this preclinical study using gene-targeted pigs with polyps [18], we could assess, for the first time, the usability of the device and demonstrate that there is no significant damage to the colon wall.

The idea behind this technology, as well as one of the further planned developments, is to implement an acoustic signal that may warn the endoscopist at the precise moment a polyp is detected thanks to the shift in dielectric properties. Hence, the endoscopist can focus his attention and concentrate on the standard endoscopic image. This constitutes a big difference compared to other existing devices that use artificial intelligence, which depict boxes on the screen [11], or side-viewing endoscopes that display the image in one or two accessory screens [24]. In other words, the technology tested in this work does not aim to provide additional visual information to the practitioner in order to avoid distractions that decrease concentration.

This microwave-based device has initially been conceived as a disposable accessory to be used as a complement to colonoscopy and, ideally, could be used in colonoscopies performed by any indication. However, similarly to what happens with other endoscopic accessories, the fact that the device has to be mounted at the tip of the colonoscope before starting the exploration adds some difficulty and can limit its use in clinical practice. Therefore, we envisage that the solution could be to integrate the antennas inside the colonoscope.

The main limitation of this study is the use of a pig model. As the anatomy of pigs differs significantly from that of humans, it is not an ideal model for assessing the feasibility of conducting a complete colonoscopy. Additionally, these genetically modified pigs have an excessive number of adenomas that are clustered, making it nearly impossible to identify them individually. Furthermore, the pigs were not administered a standard bowel preparation lavage, and although manual cleansing was performed, fecal matter remnants remained and appeared as short bright areas that may have triggered false positive detections. Lastly, only a small number of pigs were examined, which is another significant limitation. Another crucial limitation is the absence of data on relevant models for polyp detection using microwaves. The algorithm has only been finetuned with unrealistic data, such as simple phantoms or ex vivo colons. However, this study was planned in order to test security in in vivo animal models and perform a performance check.

## 5. Conclusions

This study demonstrates, for the first time, that microwave-based colonoscopy is safe and feasible and has the potential for detecting polyps in an in vivo animal model. The microwave imaging method utilized multiple antennas to capture spatial changes in dielectric tissue properties, allowing for the identification of polyps based on their contrast with healthy mucosa. The study involved experiments in healthy pigs and pigs with adenomas. In the first group, the device showed compatibility with endoscopic tools and demonstrated the ability to emit and receive microwaves without causing damage to the colon wall. With the second group, the device successfully detected polyps, but this model showed important limitations, including a high number of discarded frames due to direct contact with the mucosa or stool remnants and the clustering of polyps in the pig model. Further investigations are required to comprehensively assess the device’s efficacy in humans and the benefit of presenting the results by emitting an acoustic signal, and whether it can identify additional polyps that would have otherwise been undetected during a conventional colonoscopy should be further examined.

## Figures and Tables

**Figure 1 cancers-15-03122-f001:**
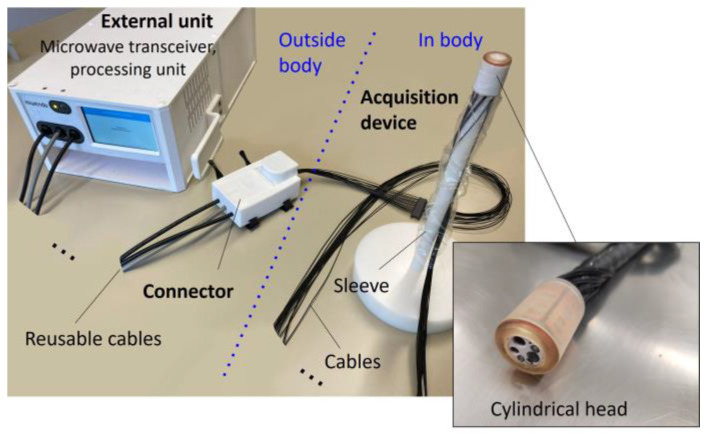
The imaging system comprises two main components: a cylindrical ring-shaped acquisition device that is attached to the colonoscope’s tip and an external unit that is connected to the acquisition device via cables. The external unit contains a microwave transceiver and a processing unit.

**Figure 2 cancers-15-03122-f002:**
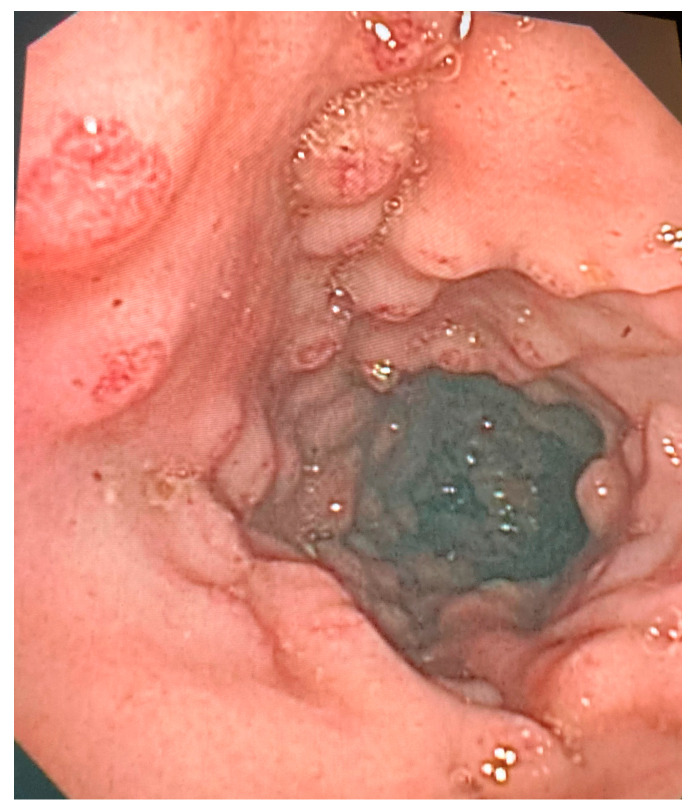
Standard endoscopic view image of a segment of the colon containing adenomatous polyps distributed in clusters.

**Figure 3 cancers-15-03122-f003:**
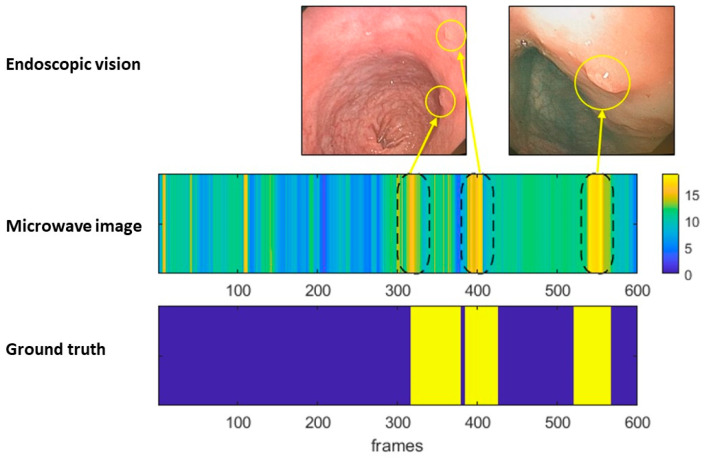
Detection in a segment containing polyps and normal mucosa as a function of time. (**Top image**): Endoscopic images of the corresponding polyps. (**Middle image**): Microwave image of the change in dielectric properties as a function of time in which regions with brighter pixels correspond to the areas with a high likelihood of containing a polyp. (**Bottom image**): Ground truth image obtained from the visualization endoscope images in which yellow bands indicate frames with the presence of polyps.

## Data Availability

The authors provide no restriction on the availability of the methods, protocols, instrumentation, and data utilized in this article. Data are available from the corresponding author upon reasonable request.

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
