# Peer review of "Preclinical Evaluation of a Microwave-Based Accessory Device for Colonoscopy in an In Vivo Porcine Model with Colorectal Polyps"

_cancers, 2023, doi:10.3390/cancers15123122_

Round 1
Reviewer 1 Report
The authors investigated the feasibility of a microwave-based colonoscopy in an in-vivo porcine model. The study identified microwave-imaging methods as feasible and safe method to detect polyps in a preclinical in vivo model. Please see below some comments and suggestions to improve the paper:
Major comments:
- Abstract: I would suggest changing the wording of the conclusions. Although the results indicate that the method is safe and feasible, it was only done in a pre-clinical model. Consider adding "in a pre-clinical model". Consider adding a future direction phrase that could test the method in a clinical trial setting.
- Discussion: What are the authors thoughts on how microwave imaging would be used in clinical practice? Would it be used complementarily with colonoscopies? Or would it even replace colonoscopies, potentially only in high-risk populations? Could you expand on the future of the clinical use of this methodology in the discussion?
Minor comments:
- Simple summary: The simple summary is supposed to be for a broad audience with potentially limited knowledge on this topic. I would suggest simplifying the language of the simple summary for a broader audience to understand the subject.
- 2.2 title: Consider rephrasing the title to "microwave imaging device" or something similar to align the wording with the text.
- Line 132: remove double period.
Author Response
Response to Reviewer 1 Comments
Major comments:
1) Abstract: I would suggest changing the wording of the conclusions. Although the results indicate that the method is safe and feasible, it was only done in a pre-clinical model. Consider adding "in a pre-clinical model". Consider adding a future direction phrase that could test the method in a clinical trial setting.
Response 1: Thank you for these suggestions that have been added to the abstract.
2) Discussion: What are the authors thoughts on how microwave imaging would be used in clinical practice? Would it be used complementarily with colonoscopies? Or would it even replace colonoscopies, potentially only in high-risk populations? Could you expand on the future of the clinical use of this methodology in the discussion?
Response 2: This microwave-based accessory device has been conceived and designed to be used as a complement to colonoscopy. Ideally, it could be used in all colonoscopies since has the potential of increasing the performance and all patients could Benefit from it. However,the fact is that all the solutions designed as accessories that have to be mounted at the tip of the colonoscop are not practical and the use is limited. Therefore, our opinion is that the future of this solution would be the integration of the antenas inside the endoscope. This has been discussed in a new paragraph (lines 307-313).
Minor comments:
3) Simple summary: The simple summary is supposed to be for a broad audience with potentially limited knowledge on this topic. I would suggest simplifying the language of the simple summary for a broader audience to understand the subject.
Response 3: We have simplified the language.
4) title: Consider rephrasing the title to "microwave imaging device" or something similar to align the wording with the text.
Response 4: The title has been rephased to be consistent with the text.
5) Line 132: remove double period.
Response 5: It has been done (line 172 in the new version).

Reviewer 2 Report
Title: Preclinical evaluation of microwave-based colonoscopy in an in 2 vivo porcine model with colorectal polyps
In this paper, the effective method for colorectal cancer prevention with several drawbacks that result in polyp miss rate has been discussed. Microwave imaging can obtain anatomical 20 and functional images of the colon since has the capacity of detecting changes on the dielectric properties of the tissues. The aim of the current study is to evaluate for the first time the feasibility of a microwave-based colonoscopy in an in-vivo porcine model. We developed a device provided with microwave antennas and was attached to a conventional endoscope. The prototype was applied in healthy pigs and it did not impair the endoscopic vision, was compatible with the endoscopic tools and deep mural injuries were not observed. When tested in gene-targeted pigs with mutations in the adenomatous polyposis coli gene, the microwave-based detection algorithm showed a maximum detection signal for the adenomas compared with the healthy mucosa, confirming its potential of detecting polyps
Few comments are given to improve the paper as
1) What is the novelty of the work, please highlight it
2) Introduction is written in a short form. Please provide more details in the introduction
3) Provide the organization of the paper at the end of introduction section
4) Why 600 frames with only 3 isolated polyps have been taken? what happens if we take greater or lessor frames or isolated polyps
5) Provide more discussion that can easy to understand for the readers
6) Conclusions needs more improvement
It is fine
Author Response
Response to Reviewer 2 comments
1)What is the novelty of the work, please highlight it.
Response 1: The novelty is the use of microwave imaging (a non-ionizing radiation) as a complement for standard colonoscopy and its first use in an in-vivo animal study. This has been highlighted in the summary and also in the introduction (lines 89 and 91) and discussion (lines 250, 287-291).
2) Introduction is written in a short form. Please provide more details in the introduction. Response 2: More details concerning the problem of CRC cancer (1st paragraph, lines 61-70), other solutions as AI for improving the performance of colonoscopy (lines 84-86) and regulatory requirements for invasive devices (4th paragraph, lines 103-110) have been provided in the introduction. We have also added two new references (1, 23).
3) Provide the organization of the paper at the end of introduction section.
Response 3: Done as required (lines 123-126).
4) Why 600 frames with only 3 isolated polyps have been taken? what happens if we take greater or less frames or isolated polyps.
Response 4: We analyzed the 600 frames that corresponded to the 3 isolated polyps. We did not have more frames from these 3 polyps and to analyze less frames could lead to losing information. All the other frames were discarded because the antennas were mismatched when they touched the wall of the colon or because they corresponded to the segments of colon with polyps distributed in clusters and, because they were located very close and without normal mucosa between them, they could not be identified individually. This information has been provided in the second paragraph of the results section (lines 203-204 and 215-225).
5) Provide more discussion that can easy to understand for the readers.
Response 5: Discussion has been changed to be more understandable for the readers and we have introduced some clinically relevant aspects that can be addressed by this novel technology (lines 269-275).
6) Conclusions need more improvement.
Response 6: Conclusions have been improved.

Reviewer 3 Report
thank you for allowing me to review this original preclinical study. the authors have evaluated the interest of a new endoscopic device allowing the detection of polyps in pigs. this study allows to demonstrate its safety in animals and its reliability to detect polyps. moreover, the authors specify very well the limits of their feasibility study. this original study needs to be confirmed in the future with other more precise experimental models.
Author Response
There are no requests
Reviewer 4 Report
Even if I do not think that this research will be able to bring a long-term benefit in the diagnosis and final treatment of patients with polyps, I think that the research deserves to be published. It will be able to open new trends in this area of pathology.
Author Response
Any request except improving the references and the conclusions. We have added two new references and changed the conclusions.
Round 2
Reviewer 1 Report
The authors addressed my comments and concerns accordingly. I have no further comments.
Reviewer 2 Report
Accept